# Nanocellulose from Cocoa Shell in Pickering Emulsions of Cocoa Butter in Water: Effect of Isolation and Concentration on Its Stability and Rheological Properties

**DOI:** 10.3390/polym15204157

**Published:** 2023-10-19

**Authors:** Catalina Gómez Hoyos, Luis David Botero, Andrea Flórez-Caro, Jorge Andrés Velásquez-Cock, Robin Zuluaga

**Affiliations:** 1Programa de Ingeniería en Nanotecnología, Universidad Pontificia Bolivariana, Circular 1 N_ 70-01, Medellín 050031, Colombia; luisd.botero@upb.edu.co (L.D.B.); andrea.florezc@upb.edu.co (A.F.-C.); jorgeandres.velasquez@upb.edu.co (J.A.V.-C.); 2Facultad de Ingeniería Agroindustrial, Universidad Pontificia Bolivariana, Circular 1 N_ 70-01, Medellín 050031, Colombia; robin.zuluaga@upb.edu.co

**Keywords:** cellulose nanofibers, cocoa shell, cocoa fat, cocoa fat-in-water Pickering emulsions, arrested coalescence

## Abstract

There is a growing interest in developing new strategies to completely or partially replace cocoa butter in food and cosmetic products due to its cost and health effects. One of these alternatives is to develop stable emulsions of cocoa butter in water. However, incorporating cocoa butter is challenging as it solidifies and forms crystals, destabilizing the emulsion through arrested coalescence. Prevention against this destabilization mechanism is significantly lower than against coalescence. In this research, the rheological properties of nanocellulose from cocoa shell, a by-product of the chocolate industry, were controlled through isolation treatments to produce nanocellulose with a higher degree of polymerization (DP) and a stronger three-dimensional network. This nanocellulose was used at concentrations of 0.7 and 1.0 wt %, to develop cocoa butter in-water Pickering emulsion using a high shear mixing technique. The emulsions remained stable for more than 15 days. Nanocellulose was characterized using attenuated total reflection–Fourier transform infrared spectroscopy (ATR–FTIR), hot water and organic extractives, atomic force microscopy (AFM), degree of polymerization (DP), and rheological analysis. Subsequently, the emulsions were characterized on days 1 and 15 after their preparation through photographs to assess their physical stability. Fluorescent and electronic microscopy, as well as rheological analysis, were used to understand the physical properties of emulsions.

## 1. Introduction

The potential applications of nanocellulose in the food and cosmetic industries have been identified since its development in the 1980s. From that point on, it has been used in oil and fat stabilization in a wide range of food and cosmetic products [1] and as a functional ingredient. In vivo tests have demonstrated that CNFs (cellulose nanofibrils) interact with glucose, limiting its diffusion [2], increasing fecal volume, absorbing harmful substances, and lowering cholesterol [3]. Nowadays, nanocellulose is not approved by the FDA (Food and Drug Administration) or EFSA (European Food Safety Authority); however, recent research suggests that it is a safe food contact material, and cytotoxic or genotoxic effects have not been identified [4,5]. There is currently a request to the FDA to include nanocellulose-based wood sources in the GRAS (Generally Recognized as Safe) list [6].

Nanocellulose has traditionally been isolated from wood and agro-industrial by-products through a multistep sequence comprising chemical or enzymatic stages, culminating in a homogenization treatment. However, the exploitation of wood-derived sources may not be convenient in tropical regions, as it could promote the degradation of native forests [7]. Therefore, agricultural by-products, such as cocoa shells, emerge as an interesting and readily available option, adding value and reducing the amount of waste produced by the agroindustry. For instance, cocoa shells represent 12 wt % of cocoa by-products and are removed either before or after roasting the cocoa nibs. In 2017, Colombia generated approximately 6.816 t of this by-product [8], which has primarily been used in animal food formulation, as fuel for boilers, and in fertilizer manufacturing [9].

In a previous study, the authors isolated a suspension of nanocellulose from cocoa shells using a four-step alkaline sequence followed by a grinding treatment. The final suspension consisted of a combination of nanocellulose, cocoa butter, and hemicelluloses, with potential applications in the food industry [10]. The presence of non-cellulosic components, such as cocoa butter, makes nanocellulose from cocoa shell an ingredient for developing oil-in-water (O/W) Pickering emulsions (PEs). In PEs, dispersed particles with sizes ranging from just a few nanometers to tens of micrometers participate in stabilizing a multi-phase structure by providing a physical barrier to emulsion droplet coalescence [11]. The physical properties of nanocelluloses, such as their high aspect ratio, wettability, and gel rheology, combined with their characteristics as functional ingredients, make this biopolymer a suitable stabilizer for food PEs. The use of nanocellulose as a stabilizer in PEs has been widely studied, with a clear indication that cellulose concentration, as well as morphological and surface properties, could be controlled to stabilize different emulsions. These emulsions are usually prepared with model non-polar phases such as dodecane [12] and with commercial liquid oils like sunflower [13], canola oil [14], and corn oil [15], among others.

On the other hand, developing PEs with crystalline fats, for instance, cocoa butter, is also relevant for the formulation of foods. Incorporating this kind of fat into emulsions is challenging, as it solidifies during its processing and/or storage and forms crystals that may protrude through the interface, leading to partial coalescence [16], a process in which two or more crystalline droplets merge to form a single irregularly shaped aggregate [17]. Stability against partial coalescence is usually lower than against coalescence [18]. They can differ by a factor of 10^6^ or even more for similar emulsions with or without crystals in the globules [18]. PEs with liquid or crystalline dispersed phases are relevant for creating foods with compositions and microstructures that can be fine-tuned to obtain the desired quality attributes in the final products [17,18]. 

Physicochemical and sensory attributes of emulsion-based food products are strongly influenced by droplet properties, such as droplet morphology, interactions, and rheological behavior [17]. There is a great deal of research attempting to link these properties in PEs [19,20,21,22]. Microscopic and rheological measurements can also be used to monitor changes in the structural organization and interactions of the droplets in emulsions. For instance, observing changes in microstructure and measuring changes in the viscosity or elastic modulus of an emulsion over time can be used to track changes in the emulsion’s physical properties and stability [23]. 

Literature about cocoa butter in water emulsions is dedicated to studying the crystallization process of cocoa butter in the emulsion [24,25]. The growing chocolate market and the significant increase in demand for cocoa butter, coupled with the limited source of cocoa butter, drive up its cost and the cost of food and cosmetic products that use it. On the other hand, consumption of chocolate is related to some health risks associated with its saturated fatty acid content (around 60%). A high-saturated fatty acid diet has been found to be related to the occurrence of high blood pressure and insulin resistance, among other issues [26,27]. Therefore, there is a growing interest in developing substitutes to partially or fully replace cocoa butter in food and cosmetic products, and these substitutes include stable emulsions. However, to the authors’ knowledge, the literature is lacking reports that evaluate the physical stability, microstructure, and rheological behavior of cocoa butter PEs, especially with products derived from biological sources such as nanocellulose.

While previous research on nanocellulose in PEs has been developed for dispersed phases that remain liquid or partially solidify, such as n-hexadecane [28], coconut oil [22], and olive oil [29], among others, they have focused on the morphology of the nanocellulose incorporated and used wood as the main cellulosic source, neglecting the influence of the isolation procedure for cellulose or alternative sources of cellulose, for instance, cocoa shells. This research is more relevant considering in the literature the use of cellulose nanofibrils to decrease oil or fat content in edible formulations, which helps to reduce the caloric value of food products while functioning as a dietary fiber, suggesting its possible use as a functional food ingredient with added health benefits [30,31,32].

For the above-mentioned objectives, an exploration of alternative sources of nanocellulose was undertaken, aiming to reduce cocoa butter consumption by evaluating the effects of nanocellulose concentration and its rheological properties in the preparation of stable O/W Pickering emulsions using cocoa butter as the dispersed phase. The development of stable O/W cocoa butter Pickering emulsions is intended to diminish the cocoa butter content in food and cosmetic formulations.

To achieve these objectives, two different chemical treatments were employed to isolate nanocellulose from cocoa shells. Differences in nanocellulose resulting from the isolation treatments were analyzed using attenuated total reflection–Fourier transform infrared spectroscopy (ATR–FTIR), extractive determination, atomic force microscopy (AFM), degree of polymerization (DP), and rheological analysis. Additionally, the physical stability of the emulsions formulated with the two types of nanocellulose was assessed through visual observation as well as fluorescence and scanning electron microscopy to investigate the morphology of the formulated Pickering emulsions. Finally, rheological analysis was conducted to examine the viscoelastic properties of both nanocellulose and Pickering emulsions.

## 2. Materials and Methods

Cocoa shells and cocoa butter, extracted from the main varieties of cocoa beans (*Theobroma cacao* L.) cultivated in Colombia, were supplied by Compañía Nacional de Chocolates (Medellín, Colombia) after the roasting process. Potassium hydroxide, hydrochloric acid, and sodium chlorite analytical grade, were purchased from Merck KGaA (Darmstadt, Germany).

### 2.1. Isolation of Nanocellulose

Cocoa shell (CS) was chemically treated according to two different methods. As described in Figure 1, C_1_ was isolated following a two-step treatment, and C_2_ was isolated using a four-step treatment developed by [10,33]. Briefly, in the case of both C_1_ and C_2_ samples, ground cocoa husk underwent an initial alkaline hydrolysis treatment with a 5 wt % KOH solution at room temperature for 14 h to remove a portion of the lignin and hemicellulose [10,33]. For the C_2_ sample, this process was followed by delignification using chlorine dioxide, produced by acidifying NaClO_2_ to pH 5. Afterward, a second alkaline treatment with a 5 wt % KOH solution was carried out for 14 h. Finally, the material underwent a reaction with 1 wt % HCl for 2 h at 80 °C [10,33]. After each step, the insoluble residue was extensively washed with distilled water until the pH was neutral. Finally, a 2 wt % suspension of the cellulosic material obtained from C_1_ and C_2_ was passed 30 times through a grinder (Masuko Sangyo, Supermasscolloider, Kawaguchi, Japan) following the G30 procedure developed by [34].

The nanocellulose suspensions C_1_ and C_2_ with concentrations of 0.1 wt % were filtered under vacuum through a 0.2 μm nylon membrane. The resulting films were oven-dried at 40 °C for 72 h and stored for further analysis. 

### 2.2. Emulsions Formulation

Cocoa butter in water Pickering emulsions (4:1) were prepared using a method based on previous works [29,35,36]. Cocoa butter was melted for one hour at 110 °C to remove all crystal memory [24]. Subsequently, as illustrated in Figure 1, nanocellulose suspension, either C_1_ or C_2_ at two different concentrations (0.7 wt % and 1.0 wt %) [22,36], was added to the molten cocoa butter at room temperature (25 °C). Finally, the two-phase system was processed through a high shear mixing (HSM) or high-speed rotor-stator system (Ultraturrax T50, IKA, Cologne, Germany, impeller model S25N-25F) at 25,000 rpm for 180 s. HSM is based on a rotor-stator system, where the rotor can rotate at 25,000 rpm and is enclosed in a stator with a gap between them ranging from 100 to 3000 µm. This equipment applies a high shear force, which disperses fat droplets in the water phase. After HSM, the samples were labeled PEC_1_ 0.7, PEC_1_ 1.0, PEC_2_ 0.7, and PEC_2_ 1.0. They were kept in a container for transportation. Table 1 describes the sample labels for PEs. 

### 2.3. Nanocellulose Characterization

#### 2.3.1. Attenuated Total Reflection–Fourier Transform Infrared Spectroscopy (ATR–FTIR)

ATR-FTIR spectroscopy analysis of C_1_ and C_2_ was performed with an FTIR spectrometer (Thermo Scientific Nicolet iS50, Waltham, MA, USA) with a single-reflection ATR and type-IIA diamond crystal mounted on tungsten carbide. The diamond ATR had a sampling area of approximately 0.5 mm^2^, and a consistent reproducible pressure was applied to every sample. The FTIR spectra were collected for 64 scans at a 4 cm^−1^ resolution. Each spectrum represents the average of three spectra, and all spectra were corrected using the advanced ATR correction of OMNIC 9, a software developed by Thermo Fisher Scientific^TM^, to eliminate the diamond crystal effect and ease the comparison with transmission spectra.

#### 2.3.2. Nanocellulose Extractives

Aqueous extractives in hot water and organic extractives in ethanol/toluene were determined for C_1_ and C_2_ using the TAPPI (Technical Association of the Pulp and Paper Industry, Atlanta, GA, USA) method T207 cm-99 (1999) and TAPPI method T204 cm-97 (2007), respectively. Soluble extractives were calculated as the difference between the weights of the dried samples before and after the extraction process.

#### 2.3.3. Atomic Force Microscopy (AFM)

The morphology of nanocellulose fibers in C_1_ and C_2_ was evaluated by AFM. Samples were imaged in tapping mode with a Flex AFM (Nanosurf, Liestal, Switzerland) equipped with a multimode head and operated with a resonance frequency of 183 kHz. It utilized a cantilever of 125 mm in length and a tip radius of 5–10 nm. Nanocellulose samples were diluted in distilled water and sonicated using an Elma Elmansonic P ultrasonic bath sonicator operating at 37 Hz and 70% power for 15 min at room temperature to achieve a good dispersion of the nanocellulose without introducing significant morphological changes. Subsequently, a fine layer of the sample was deposited on a mica substrate. Before the morphological analysis, samples were stored in a vacuum desiccator for 3 days. 

#### 2.3.4. Degree of Polymerization (DP)

The degree of polymerization (DP) of C_1_ and C_2_ was determined according to ISO (International Organization for Standardization, Geneva, Switzerland) 5351-2010 [37]. Dried samples were torn into fine pieces and were introduced into an amber flask with 5 copper rods and 25 mL of distilled water and stirred. After the dispersion of the specimens, 25 mL of the CED solution (1 mol L^−1^) was added. A flow of nitrogen was used to expel the air inside the flasks. Samples were mixed until the cellulose had completely dissolved. The solutions were then introduced in a controlled-temperature bath and measured in a calibrated capillary viscometer (Lagge, LV0003) at 25 °C [34]. The time required to elute the solution was measured. From this information, the DP of the samples was calculated according to the Staudinger-Mark-Houwink equation for a cellulose-CED solution, expressed in Equation (1).
(1)$$\left[\eta \right]={Κ}_{DP}DP^{\alpha }$$
$\left[\eta \right]$ is the intrinsic viscosity of the solution (units mL/g), *DP* is the degree of polymerization of the cellulosic material (dimensionless), and K*_DP_* (units mL/g) and α  (dimensionless) are experimental constants defined by Marx Figini [38] for a cellulose-CED solution. These constants depend on the *DP* of the cellulosic material and are indicated elsewhere.

#### 2.3.5. Rheological Analysis

Rheological measurements of nanocellulose were conducted using a Discovery HR-2^®^ rheometer (T.A. Instruments, New Castle, DE, USA) equipped with a 40 mm plate-plate geometry and a fixed gap between the plates of 0.5 mm [39]. All measurements were performed at 20 °C, following an equilibration time of 10 min [22,40]. A solvent trap was used to prevent water evaporation [41]. Amplitude viscoelastic measurements were performed using a frequency of 6.28 rad/s and an amplitude sweep between 0.5 Pa and 1000 Pa. A frequency sweep was carried out from 6.28 to 62.8 rads/s at a constant shear stress within the linear viscoelastic region of each sample. The values of the energy storage (G′) and loss moduli (G″) were recorded. Flow curves were obtained by measuring viscosity during logarithmic increases in shear stress from 0.1 to 100 rad/s.

### 2.4. Emulsion Characterization

#### 2.4.1. Physical Stability

The stability of all the emulsions was evaluated by visual observation with photographs taken 1 and 15 days after preparation.

#### 2.4.2. Fluorescent Microscopy

The distribution of fat and cellulosic phases in nanocellulose and Pickering emulsions was examined by fluorescence microscopy (Zeiss Axio Observer, Zeiss, Oberkochen, Germany) with 10× and 40× objectives. In the sample preparation, 1 mL of nanocellulose was stained with 8 μL of Nile Red (for the oil phase) and then with 6 μL of Calcofluor White (for the nanocellulose); 4 μL of the prepared samples were used for this study [42]. Samples were covered using a glass coverslip (18 × 18 × 2 mm^3^), and the borders were protected with nail polish to fix the borders of the coverslip and prevent evaporation [42,43].

#### 2.4.3. Scanning Electron Microscopy

The morphological features of the Pickering emulsions were examined by scanning electron microscopy (SEM) with a JEOL JCM-6000 (Akishima, Japan) scanning electron microscope at an acceleration voltage of 15 kV. Before the analysis, a drop of each sample was placed on carbon tape and evaporated at room temperature. Subsequently, the samples were coated with gold.

#### 2.4.4. Rheological Analysis

Amplitude viscoelastic measurements and frequency sweeps were conducted on emulsions following the same procedure used in nanocellulose, described in Section 2.3.5. 

#### 2.4.5. Statistical Analysis

Data for this study were obtained through a series of experiments aimed at characterizing Pickering emulsions. The primary variables of interest included isolation, treatment, and concentration. Experiments were conducted in triplicate to ensure data reproducibility. All data preprocessing procedures were conducted using software specific to each technique. 

Descriptive statistics were employed to provide a summary of the collected data, including measures such as mean and standard deviation. To aid in the interpretation of results, graphical representations were created using Origin. Scatter plots, box plots, and bar charts were utilized to visualize trends and relationships within the data.

## 3. Results and Discussion

### 3.1. Physicochemical Characterization of CNFs: C_1_ and C_2_

CNFs isolated from cocoa shells will undergo evaluation for the development of cocoa butter-in-water Pickering emulsions. These emulsions stabilize as the CNFs create a physical barrier and a network, acting as an obstacle between contiguous droplets and preventing droplet coalescence [44]. Factors influencing the stability of such emulsions include the morphology, size, and shape of the CNFs, as well as their chemical composition, which can affect the surface charge and interactions. The morphology and chemical composition of CNFs are closely linked to their isolation process [44]. Hence, an assessment was conducted on the chemical, physical, and morphological characteristics of both CNFs, C_1_ and C_2_, to comprehend the impact of the isolation process on the physical barrier and network responsible for stabilizing the Pickering emulsion.

Surface hydroxyl groups in CNFs make them hydrophilic polymers capable of stabilizing oil-in-water (O/W) emulsions [44]. ATR-FTIR is a commonly used technique for studying these groups and their interactions with both cellulosic and non-cellulosic components present in CNFs [34]. The ATR–FTIR spectra of C_1_ and C_2_ films are presented in Figure 2a,b. They exhibit the characteristic peaks of cellulose at 3335 cm^−1^ and 3285 cm^−1^, assigned to hydrogen bonds, and at 711 cm^−1^, assigned to I_β_ cellulose [45]. The spectra also evidenced the presence of non-cellulosic components, such as hemicelluloses, lignin, and fat, as observed by the peaks present at 2953, 1462, and 1377 cm^−1^, related to cocoa butter structures [46,47], while the bands at 1743, 1590, 1510, and 1235 cm^−1^ are related to residual hemicelluloses and lignin [45]. These results agree with previous works on the isolation of nanocellulose from cocoa shells [10].

Hemicelluloses are one of the most difficult components to remove from cell walls owing to their strong interaction with cellulose and lignin [33,34,48]. It could be physically interwoven into the cell wall through the formation of hydrogen bonds, physically bond to cellulose [33,48], or form covalent complexes with lignin by ester bonds [49]. Hemicelluloses were liberated from cocoa shells by alkaline extraction [33], which can cleave ester bonds between lignin and hemicelluloses in the cell walls, freeing a fraction of the lignin present [49]. It has been demonstrated that the extraction of hemicellulose from a partially delignified material results in a more efficient process [50], as is the case of the C_2_ sample, which includes two alkaline reactions with 5 wt % KOH for 14 h, one before and the second one after delignification with NaClO_2_ (as presented in Figure 1), while C_1_ includes only one reaction with 5 wt % KOH for 14 h, before delignification.

As mentioned, other non-cellulosic components were identified in the FTIR spectra, such as cocoa butter from the cocoa shell. Both cellulosic samples C_1_ and C_2_ exhibit several vibrations related to functional groups of the main components of this fat, such as bands associated with palmitic and capric fatty acids (2953 cm^−1^, 2922 cm^−1^, 2853 cm^−1^, 1462 cm^−1^, 1377 cm^−1^) [10,46,47,51]. The absorbances of these vibrations are lower in C_2_, as the extraction of cell wall polymers, e.g., pectic polysaccharides and hemicellulose, increases the extraction of oil or fat from the lignocellulosic material [52,53,54]. The presence of vegetable fat was also evidenced by fluorescent microscopy (Appendix A). Despite the hydrophilic character of cellulose, the fluorescence images show the coexistence of fat particles (red) and cellulosic material (blue) in a homogeneous suspension, with no evidence of phase separation despite the nature of both components [10]. 

The presence of non-cellulosic components in CS, C_1_, and C_2_ was also verified by soluble extractive tests in hot water and organic solvents presented in Table 2. Results show that CS and C_1_ have a higher percentage of both aqueous and organic extractives. The organic solvent-extractable material in the samples is composed of fatty acids, resin acids, waxes, and tannins [55]. For this reason, absorbances at 2953, 2922, and 2853 cm^−1^ associated with asymmetric stretching of aliphatic –CH_3_ and symmetric stretching of aliphatic –CH_2_ in fatty acids are significantly reduced in CS, C_1_, and C_2_ samples after extraction with organic and aqueous solvents (CS_E, C_1__E, and C_2__E in the FTIR-ATR spectrum, Figure 2). The hot water-soluble part of the untreated material consists primarily of low-molecular-weight molecules, such as carbohydrates, inorganic salts, polyphenols, sugars, and other water-soluble components [56].

Finally, changes after the different isolation treatments were evaluated by comparing three regions in the FTIR-ATR spectrum, from 3600 to 3000 cm^−1^, which are associated with the OH stretching frequencies of cellulose, correlating with the hydrogen bonding pattern [57,58,59]. This vibration range exhibits poor resolution. However, mathematical processing such as the deconvolution of spectra obtained in FTIR spectrometers can be used to enhance the resolution [60]. Appendix A displays the deconvolution of all FTIR spectra between 3000–3650 cm^−1^. The height of the peaks at 3405 cm^−1^, 3305 cm^−1^, and 3285 cm^−1^, which related to the O(2)H⋯O(5) intermolecular bonding [59,61], O(6)H⋯O(3) intermolecular bonding [45], and OH stretching intermolecular hydrogen bonds in the 101 plane, respectively, increased in C_1_ with respect to C_2,_ which is associated with a higher number of intermolecular hydrogen bonds in C_1_. Changes in hydrogen bonding associated with different isolation processes have been reported by other authors [34]. Table 3 presents the absorbances and descriptions of the vibrations related to the hydrogen bonds.

Previous studies on banana rachis and cocoa husk have proposed the use of an alkaline four sequence to isolate C_2_ CNFs [10,33], as described in Figure 1. This sequence is based on the cleavage of β-aryl-O-4 linkages in lignin by alkaline media, exposing new zones for the following treatment [63], the degradation of phenolic and non-phenolic structures in lignin by chloride dioxide [64]. After the elimination of lignin, a secondary alkaline treatment can react with muconic acids produced during delignification while using the freshly exposed surfaces to cleave the hemicellulose-cellulose bonds present, reducing the presence of hemicelluloses in the final structure [65]. Finally, a demineralization treatment with HCl at a low concentration was used; the acid concentration allowed a reduction of the cellulosic hydrolysis [33]. Therefore, the influence of a second alkaline treatment, accompanied by an HCl treatment in C_2_, favored the removal of non-cellulosic components, resulting in a greater number of individual fibers and smaller diameters after the mechanical treatment [33,66,67], as can be observed in the AFM images presented in Figure 3.

Consequently, the isolation treatment influences the chemical composition of cellulose and its fibrillation [68]. Figure 3a–f shows that both samples C_1_ and C_2_ were nanofibrillated and presented an entangled network of cellulosic material. The formation of this three-dimensional entangled network is relevant in PE formulation because it favors the irreversible adsorption of nanocelluloses at the oil-water interface, increasing the steric hindrance between the droplets and inhibiting the free movement of the droplets, resulting in a lower coalescence rate [69].

For this reason, differences between the entangled networks formed by C_1_ and C_2_ should be comprehended with the aim of understanding the properties of PEs. Figure 3a–c present the morphology of C_1_ at different scales. AFM images showed individual nanofibers with diameters around 30.5 ± 12.5 nm and the presence of fiber bundles and sub-fibrillated cellulosic material with diameters ranging between 275 and 380 nm. In addition, Figure 3d–f shows the morphology of C_2_. It is evident that there are separate individual nanofibers with diameters around 26.9 ± 8.2 nm and fiber bundles with diameters ranging between 98 and 225 nm. AFM images reveal the presence of bundles in both samples C_1_ and C_2_, marked with white arrows in Figure 3b,c and Figure 3e,f, respectively. In summary, AFM revealed that C_1_ shows an entangled network with a lower degree of fibrillation, which is associated with its isolation treatment [34]. C_1_ CNFs also showed the presence of hemicellulose, which is related to a lower degree of fibrillation and bundles with larger diameters [68].

On the other hand, the isolation process influences the length of CNFs [34,70]. Zuluaga et al. evidenced that a reaction with HCl 2 M longitudinally cuts the cellulose microfibrils. This is typical of CNFs treated with strong acids that preferentially degrade the disordered regions along the microfibrils [33]. The length and polymerization degree of CNFs are interconnected parameters that cannot be changed independently easily [68]. A higher polymerization degree (DP) is directly related to longer fibers [39,41,43], and fiber length is a parameter that directly affects the stability of PEs, allowing the formation of a three-dimensional network that would enclose the fat droplets [71]. Also, long fiber aggregates form an entangled network, increasing the viscosity of the formed suspension compared to fibers with a lower aspect ratio [41,72]. Aiming to improve the understanding and explain the behavior of CNFs as a Pickering emulsifier, the DP of the nanocellulose obtained by the different treatments was assessed.

In CNFs, the DP is measured through its intrinsic viscosity, which is a characteristic of macromolecules that is directly related to their ability to disturb the flow and indirectly relates to the chain length. The relation between intrinsic viscosity and polymerization degree for cellulose was established in the 20th century by different authors [38], and the constants used by Marx-Figini were used to determine the DP reported in Table 1 [38]. The DP obtained for C_1_ and C_2_ was 1098.52 ± 1.10 and 554.31 ± 1.67, respectively, which are within the range reported in the literature [34,73,74,75]. The DP of a cellulosic polymer can be around 10,000 in wood cellulose, but after its degradation and purification process, the DP is reduced to about 300–1700 [73]. In cellulose from agroindustrial by-products like banana rachis, DP was reported between 1012 and 612, depending on the homogenization process [34]. Another aspect that influences the DP is the presence of non-cellulosic components. It was reported that a higher concentration of non-cellulosic components is related to a higher DP, as was observed for commercial cellulose [74] and cotton cellulose [75]. Finally, HCl used to remove mineral traces in C_2_ [33] also reduces the polymerization degree, as it degrades the short, disordered, non-crystalline regions in cellulose [75,76]. Therefore, the higher DP observed for C_1_ is clearly associated with a higher presence of non-cellulosic components and a lack of mild acid hydrolysis present in the C_2_ sample.

As previously noted, the assembly of long fiber aggregates can create a complex three-dimensional network, resulting in increased suspension viscosity when contrasted with fibers possessing a smaller aspect ratio [41,72]. Consequently, an increase in the degree of polymerization (DP) and intrinsic viscosity (Table 2) may be linked to changes in their shear flow behavior, as shown in Figure 4a, for both samples. To assess the rheological properties of the cellulosic suspensions, evaluations were conducted at two different concentrations: 0.7 and 1.0 wt % for both C_1_ and C_2_. Shear flow and oscillatory rheology results are presented in Figure 4a,b.

The shear flow behavior of the cellulosic samples is depicted in Figure 4a. The apparent viscosity of C_1_ and C_2_ decreased with increasing shear rate; this behavior is typical of non-Newtonian fluids such as shear-thinning fluids, which is the expected behavior for suspensions or fibers since these tend to orient themselves and partly disentangle due to the hydrodynamic forces exerted over them, resulting in a lower viscosity as the shear rate increases [77]. The floc breakage corresponds to the three regions observed in Figure 4a. In the first region, there is a marked shear-thinning behavior, with a decrease in sample viscosity at higher shear rates. It is followed by a plateau around 10 s^−1^, where the formation of a more entangled network reduces the floc breakage and maintains the viscosity at a stable value. Finally, the shear rate is high enough to fracture this structure, and a further reduction in the viscosity occurs as the entangled network releases the individual fibers [72].

The viscoelastic characterization of C_1_ and C_2_ shows the behavior of G′ and G″ in the linear viscoelastic region in Figure 4b. They are associated with the strength of the entangled network. It can be observed that G′ and G″ depend on the type of sample used, whether it is C_1_ or C_2_, and their respective concentrations. Comparing G′ and G″ for C_1_ and C_2_ at the same concentration, it is observed that both modules are one order of magnitude higher for C_1_ compared to C_2_, probably because of differences in fibrillation degree, as observed by AFM. It has been reported that fibrillation influences network strength; higher fibrillation of CNFs will result in lower network strength and lower values for G′ and G″, as is the case with C_2_ [41]. Saarinen et al. showed, using a plate-plate geometry, that the storage modulus of mechanically disintegrated cellulose suspensions decreases when the level of fibrillation is increased in a microfluidizer [78].

On the other hand, fiber length plays a role in viscoelastic properties. Longer fibers are expected to have higher G′ and G″, as was the case with C_1_, whose higher DP was related to longer fibers. Pääkkö et al. reported that at a concentration of 2% *w*/*w* of nanocellulose, it leads to G′ ≈ 10^3^ Pa, whereas 2 wt % of modified rodlike cellulose crystallites leads to G′ ≈ 10^1^ Pa [79]. The higher elastic modulus is explained by longer fibrils, and fibril aggregates in CNFs can form an inherently entangled network structure [79]. In addition, more flocs composed of longer and thicker fibers increase the elastic contribution [78].

For this reason, the viscoelastic properties of nanocellulose were evaluated. Linear viscoelastic properties of C_1_ 0.7 wt %, C_1_ 1.0 wt %, and C_2_ 1.0 wt %, exhibit gel-like properties (G′ > G″), as has been reported for different nanocellulose suspensions. The storage modulus G′ was higher than the loss modulus G″ in a range of frequencies without a crossover. However, C_2_ 0.7 wt %, exhibits gel-like behavior at lower frequencies. When the angular frequency increases beyond 40 rad/s, it exhibits a cross-over point at which the system shows equal contributions from both elastic and viscous components. After that point, higher frequencies weaken the structure, and C_2_ behaves as a viscose fluid, where G″ > G′ [41,72,79,80,81,82,83,84,85,86]. In C_1_ and C_2_, as the concentration of nanocellulose increases from 0.7 to 1.0 wt %, a stronger fibrous network is formed, resulting in an increase in the dynamic moduli G′ and G″. Finally, no statistical differences were observed between G′ and G″ for C_1_ at 0.7 wt % and C_2_ at 1 wt %. This means that at those concentrations, both samples form a network with the same rheological characteristics.

AFM, fluorescence microscopy, and rheological analysis of CNFs indicated that C_1_ and C_2_ form a strong three-dimensional network [22] with a gel-like structure that limits the flow of cocoa butter droplets through the water phase [87], as evidenced in fluorescence microscopy. However, as observed through the characterization of both types of CNFs, there are differences in fibrillation morphology, DP, and rheological properties between both types of nanocellulose, which could influence their emulsifying capacities [71]. To assess the extent of this contribution, a qualitative stability assay was performed. 

### 3.2. Physicochemical Characterization of Pickering Emulsions Stabilized with Different Concentrations of CNFs: PEC_1_ 0.7, PEC_1_ 1.0, PEC_2_ 0.7, and PEC_2_ 1.0

The photographs of the physical stability test for cocoa butter suspensions are presented in Figure 5a–c. PEC_2_ 0.7 creamed on day 0 (dashed rectangles in Figure 5a). Creaming was not evident in PEC_2_ 1.0; however, the emulsion was not stable over time, and the formation of fat-coalesced droplets of around several millimeters can be observed after 15 days of storage (white arrows in Figure 5c). During the assessed period, separation of cocoa butter was not evident for PEs prepared with the CNFs that exhibit higher viscosity and a stronger three-dimensional network in linear viscoelastic properties, PEC_1_ 0.7 and PEC_1_ 1.0, indicating that there was no emulsion breakage.

To the authors’ knowledge, the development of a stable cocoa butter-in-water Pickering emulsion has not been reported in the literature and is an interesting topic for the cosmetic industry, where cocoa butter is used as a moisturizer and emollient in product formulations [88]. The formation of a stable suspension of cellulose and cocoa butter is an interesting phenomenon because oil-in-water (o/w) emulsions, where the oil droplets have at least partly crystallized but the continuous aqueous phase remains liquid, are frequently less physically stable than the same systems where the oil droplets are liquid [89]. According to the literature, crystallization of the fat structures in cocoa butter-in-water emulsions occurs during their storage and causes destabilization [24,25]. 

The mechanism of physical instability for crystalline or partially crystalline lipids in o/w emulsions is arrested coalescence [89], which arises when the crystals of fat penetrate the interphase layer and, following a collision between droplets, penetrating the layer of a second droplet, allowing lipid-to-lipid contact [18,90]. The liquid oil flows out from both droplets to reinforce the link, but the fat crystals do not form spherical droplets; they maintain the shape of the individual droplets, as can be observed in Figure 5c, where some non-spherical particles can be observed on the surface of the glass container. 

To observe the changes exerted on the microstructure of the emulsion at different times and to identify them, a fluorescence microscopy analysis was performed as described by Bai et al. in the sample at days 0 and 15, after processing them, and they are depicted in Figure 6 [43]. Oil droplets are dyed with Nile red and are observed as red objects, while Calcofluor white was used to dye the cellulosic structures and are seen as blue objects in the image. Samples formulated with C_2_ (PEC_2_ 0.7 and PEC_2_ 1.0) did not show satisfactory stability before 15 days, as droplet agglomerates were observable to the naked eye, as marked with white arrows in Figure 5c. As their size was larger than the microscope focus, these images were not included in Figure 6.

Fluorescence microscopy images evidence that cocoa butter droplets in the emulsion can be divided into a fine dispersion of oil droplets embedded in a cellulosic network and a rough distribution of bigger coalesced fat droplets. The formation of nonspherical coalesced particles associated with arrested coalescence is evident in all emulsion samples since Day 0, as indicated by the arrows in Figure 6a–d. In addition, as observed in the physical stability test presented in Figure 5, PEC_1_ 0.7 and PEC_1_ 1.0 did not evidence extensive droplet destabilization during the assessed period (day 0 to day 15), because most of the cocoa butter is entrapped in the flocs formed by the CNFs network. Despite the same concentrations being used to formulate the emulsions with C2 CNFs, it is evident that in this case, the CNF network (blue) does not cover each cocoa butter droplet (red) and therefore does not form a dense barrier of CNFs to prevent coalescence [91]. Other authors have reported that this concentration depends on the different physicochemical characteristics of the cellulosic material as well as on the oil and aqueous phases [42], as observed in emulsions obtained with nanocellulose [22].

To further investigate the surface coverage, fat droplets were characterized by SEM after water evaporation at room temperature, as seen in Figure 7. SEM images revealed that nanofibrillated C_1_ and C_2_ formed a covering layer via interfacial adsorption (marked with arrows in Figure 7), as reported by other authors [71]. However, according to Manning et al., when fat crystals grow under physical restriction, a smooth surface morphology is observed, but without such restriction, crystals grow multidirectionally (as marked with dashed circles in Figure 7). In this case, nanocellulose in cocoa butter acts as the physical barrier that restricts crystal growth. SEM revealed that all emulsions present zones where multidirectional crystals can be observed, with a morphology very similar to that observed by Manning & Dimick (1985), marked by dashed circles in Figure 7 [92]. This kind of morphology is more pronounced in PEC_2_ 0.7 and PEC_2_ 1.0 than in PEC_1_ 0.7 and PEC_1_ 1.0 emulsions, which confirms that in those samples, CNFs did not cover each cocoa butter droplet. 

The rheological properties of the emulsion are powerful techniques to complement the interpretation of physical stability and interfacial interaction between cellulosic fibers and cocoa butter droplets observed in microscopy. Linear viscoelastic properties were assessed for the stable emulsions 1 and 15 days after processing (Figure 8). Results for the viscoelastic properties of the emulsion allowed us to establish that PEC_1_ 0.7 and PEC_1_ 1.0 can be classified as gel-like emulsions, as the storage modulus G′ showed a higher value than the loss modulus G″, and G′ is independent of frequency [22,93,94]. The elastic behavior dominates over the viscous one for the evaluated PEs, as the value of G′ is greater than G″ through the evaluated frequency range [22,93].

The independence of G′ from frequency is characteristic of an elastic behavior where the material’s ability to store and return energy does not vary with the rate of deformation. It is noticeable that the structure of gels has been described as a network of interacting rheological units, with G′ depending on the strength and number of interactions [95]. The formation of a gel network has been associated with enhanced creaming stability in casein-stabilized systems [96], while emulsion gels with high mechanical properties, such as storage modulus, exhibit higher chemical stability in their dispersed phase, suggesting a physical role in the stability of these structures [97]. Also, due to their firmness and stability, gels can provide a desirable texture and mouthfeel to food products. They can add creaminess, thickness, and smoothness to various foodstuffs [23]. 

As observed in microscopic (Figure 5) and SEM images (Figure 7), CNFs formed a three-dimensional network around droplets after high shear mixing, which plays a prominent role in stability and rheological behavior [98]. As observed in Figure 7a–d, nanocellulose was absorbed over the surface of the oil droplets to form a covering layer after the high shear mixing processing, resulting in a stable droplet-fiber 3D network structure. This network structure contributes to the elasticity of the emulsion. CNF networks can deform elastically when subjected to stress and then return to their original shape when the stress is removed. 

PEC_1_ emulsions stabilized by nanocellulose with different dosages showed a significant difference in elastic modulus. For example, at a shear rate of 10 rad/s, the G′ of PEC_1_ 0.7 and PEC_1_ 1.0 were 0.0750 MPa and 0.0546 MPa, respectively, indicating that the emulsion stabilized by a higher CNF concentration exhibited stronger gel behavior. This increment in G′ at higher concentrations of CNFs is consistent with previous reports, where CNFs from agro-industrial by-products were used to stabilize PEs. 

Velásquez-Cock et al. extracted CNFs from banana rachis following the same procedure as that used for C_2_ and evaluated the effect of CNF concentration on the rheological properties of coconut or palm oil-in-water-based Pickering emulsions [22]. They reported an increase in G′ values around three orders of magnitude when CNF concentration increased from 0.15 wt % to 0.7 wt %. Another study added CNFs from banana rachis, isolated using C_2_ methodology, to Curcuma longa suspensions and reported similar results. Increasing the CNF concentration from 0.1 wt % of CNFs to 0.9 wt % enhanced G′ by nearly 3 orders of magnitude and improved the stability over 30 days [93]. Banana rachis and cocoa husk are both agro-industrial by-products. Despite using the same CNF isolation methodology, the resulting CNFs exhibit different physicochemical characteristics, mainly due to variations in their physicochemical properties [10,33]. These differences necessitated different concentrations for stabilizing PEs, as is the case for C_1_ and C_2_ CNFs. 

Finally, the observed increase in G′ and G″ after storage corresponds to an enhancement of gel strength in CNFs [99]. As mentioned, the crystallization of cocoa butter-in-water emulsions occurs during emulsion storage and destabilizes the suspension by enhancing particle coalescence [24,25]. Cocoa butter is a complex fat known to crystallize in six different polymorphs [100]. These various forms of cocoa butter crystallization can significantly impact the properties of emulsions formulated with this ingredient. For example, the different crystalline forms have distinct melting points, with the α polymorph being less stable than the β form, which typically has a higher melting point [101]. Therefore, if cocoa butter in an emulsion undergoes a phase transition from a more stable to a less stable form during storage or processing, it can lead to undesirable changes in texture, appearance, and product deterioration [101]. Furthermore, the crystallization of solid fats plays a pivotal role in the partial coalescence of droplets in oil-in-water (O/W) emulsions [101]. This type of destabilization is of particular interest in food applications and is a prerequisite for producing whipped cream and ice cream [101,102]. Additionally, the crystallization behavior of these fats in emulsions is intricate and subject to various influences, such as droplet size, collision dynamics, additives, emulsifiers, crystallization medium, and thermal fluctuations during storage [101,103]. It has also been reported that shear forces can impact the crystallization of fats in emulsions and promote the formation of higher polymorphs [104].

In the cases of PEC_1_ 0.7 and PEC_1_ 1.0, cocoa butter crystallization did not destabilize the emulsion. To the author’s knowledge, there are no literature reports of stable cocoa butter in water emulsions. Hindle et al. developed cocoa butter in water emulsions using 0.8% *v*/*v* Tween 20 or 1.0 wt % caseinate, but this research was focused on droplet collision-mediated heterogeneous nucleation and its effect on the crystallization kinetics of cocoa butter [24,25]. Therefore, the findings of this work are significant from a scientific perspective and have implications for various applications, including food, cosmetics, and pharmaceutical formulation. Some potential directions and applications stemming from this research are as follows:The use of cocoa butter in water Pickering emulsions in the food industry offers several key advantages: reduced caloric density of products [105], the potential for incorporating active ingredients to enhance nutritional value [106], and lowered production costs due to reduced raw material usage [100]. However, further research is required in this field, particularly concerning the crystallization of cocoa butter with the integration of new active compounds.The formulation of innovative aqueous-based products, such as nano-delivery systems and edible coatings, can be developed using stable cocoa butter-in-water emulsions. This approach leverages the antioxidant, moisturizing, and barrier properties of cocoa butter, which is widely utilized in the food, pharmaceutical, and cosmetic industries [107].

## 4. Conclusions

In this research, an isolation treatment was used to control the rheological properties of nanocellulose from cocoa shells. Subsequently, the influence of the rheological behavior of nanocellulose on the stabilization of cocoa butter in a water Pickering emulsion was evaluated. The ATR-FTIR spectra, in conjunction with the extractives in hot water and organic solvents, indicated the partial elimination of non-cellulosic components for both samples. However, a higher presence of non-cellulosic components was observed in C_1_, in addition to a higher number of hydrogen bonds. This difference in chemical composition is attributed to the influence of a second alkaline treatment accompanied by a treatment with HCl in C_2_. These reactions also favored the breaking of hydrogen bonds. In addition, according to AFM and rheological analysis, both samples form a strong entangled network of nanofibers with a gel-like structure that limits the flow of the cocoa butter droplets in the aqueous phase. However, the second alkaline and HCl reaction promotes a higher degree of fibrillation of the material, resulting in nanocellulose with smaller diameters, more individually separated, and a lower degree of polymerization. This results in lower network strength, as evidenced by lower G′ and G″ values and lower apparent viscosity in C_2_. 

Thus, C_1_ presented higher DP and a stronger entangled network, which improved the stabilization of the cocoa butter in water-based Pickering emulsions, as analyzed by physical stability visualization and rheological analysis. C_1_ showed an enhanced 3D droplet-fiber network structure. Fat crystallization caused arrested coalescence but did not destabilize the PEC_1_ 0.7 and PEC_1_ 1.0 emulsions, even during 15 days of storage. Emulsions developed in this research offer the possibility of reducing the content of cocoa butter in food and cosmetic formulations.

## Figures and Tables

**Figure 1 polymers-15-04157-f001:**
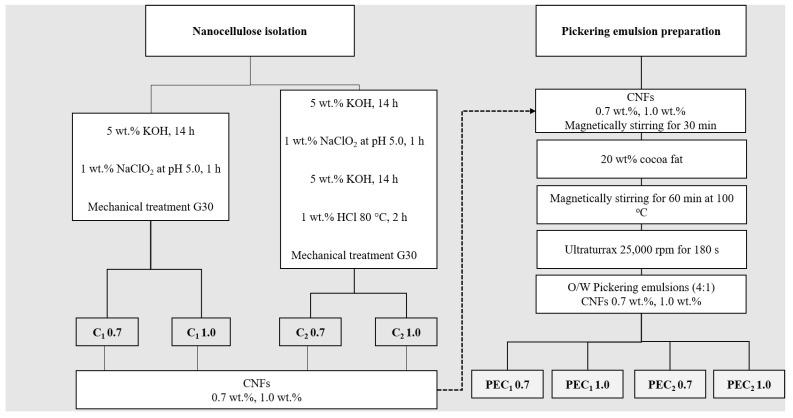
Scheme illustrating the process of isolation and preparation of nanocelluloses and their Pickering emulsions. C_1_ 0.7, C_1_ 1.0, C_2_ 0.7, and C_2_ 1.0 represent the CNFs isolated through treatments C_1_ and C_2_, with 0.7 and 1.0 indicating the CNF concentrations. PEC_1_ 0.7, PEC_1_ 1.0, PEC_2_ 0.7, and PEC_2_ 1.0 denote the Pickering emulsions stabilized by CNFs, with 0.7 and 1.0 specifying the CNF concentrations used in the PE formulation.

**Figure 2 polymers-15-04157-f002:**
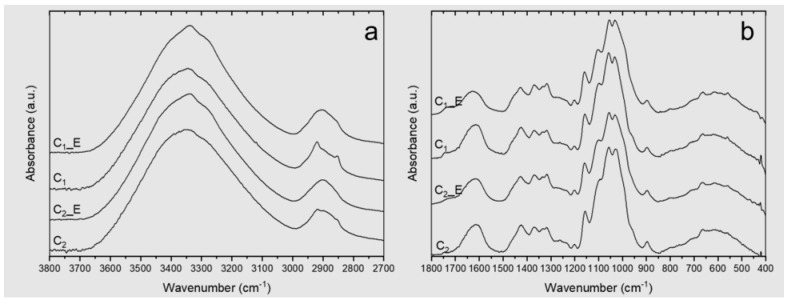
ATR–FTIR spectra of the C_1_ and C_2_ films. _E represents the sample after removing their extractives: (**a**) ATR-FTIR spectrum between 3800 and 2700 cm^−1^; (**b**) ATR-FTIR spectrum between 1800 and 400 cm^−1^.

**Figure 3 polymers-15-04157-f003:**
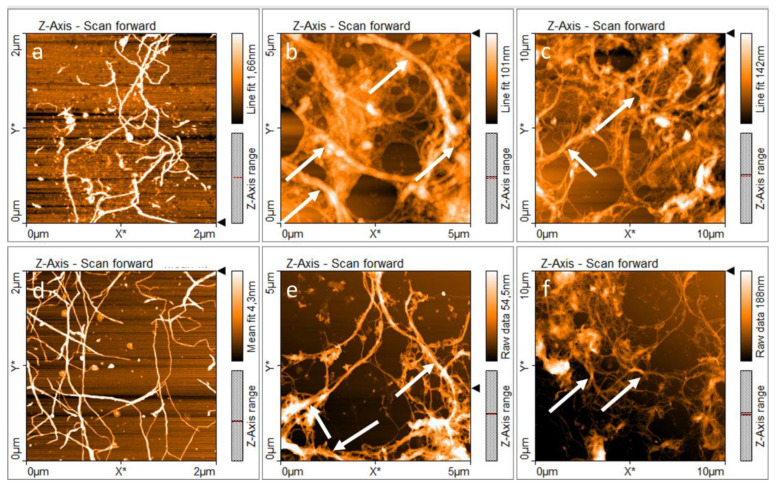
AFM images of nanocellulose: (**a**–**c**) C_1_ and (**d**–**f**) C_2_. White arrows indicate the presence of bundles. White arrows indicate the CNFs bundles.

**Figure 4 polymers-15-04157-f004:**
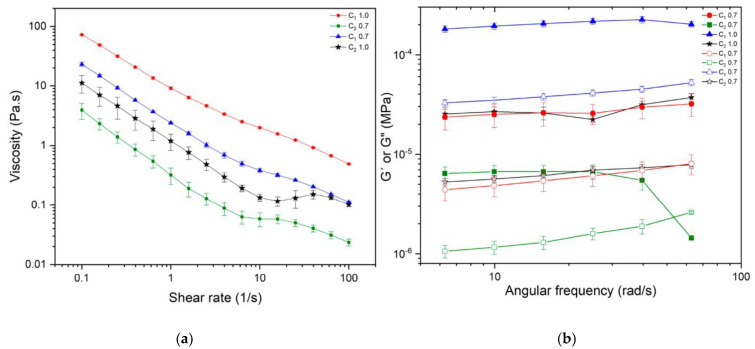
Rheological characterization of CNFs isolated from cocoa husks C_1_ and C_2_: (**a**) viscosity curves and (**b**) frequency sweep, with G′ represented by filled symbols and G″ by empty symbols.

**Figure 5 polymers-15-04157-f005:**
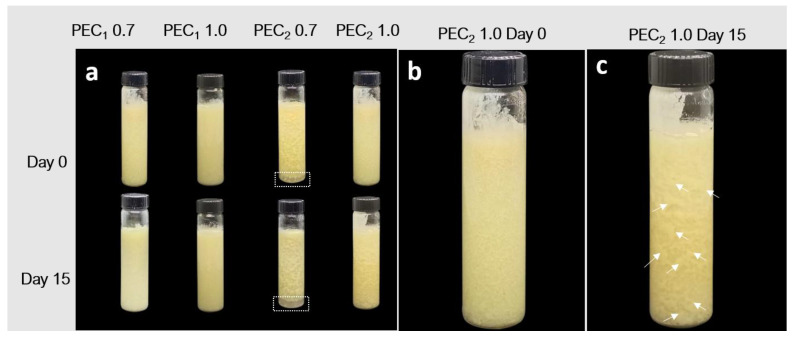
Photographs of cocoa butter Pickering emulsions: (**a**) visual appearance of PEC_1_ 0.7, PEC_1_ 1.0, PEC_2_ 0.7, and PEC_2_ 1.0 samples at days 0 and 15 of storage; (**b**) visual appearance of PEC_2_ 1.0 at day 0 of storage without evident creaming; (**c**) visual appearance of PEC_2_ 1.0 at day 15 of storage, showing fat-coalesced droplets pointed out by white arrows in the image, indicating emulsion destabilization. PEC_2_ 0.7 creamed on day 0 as indicated by dashed rectangles in Figure 5a.

**Figure 6 polymers-15-04157-f006:**
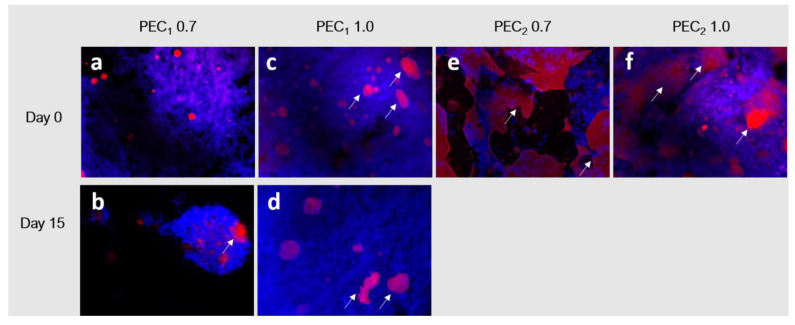
Fluorescent microscopy images of PEC_1_ 0.7, PEC_1_ 1.0, PEC_2_ 0.7, and PEC_2_ 1.0 samples at (**a**,**c**,**e**,**f**) day 0 and (**b**,**d**) day 15 of storage. Oil phase is shown in red, and the cellulose is shown in blue. The white arrows indicate the presence of fat particles.

**Figure 7 polymers-15-04157-f007:**
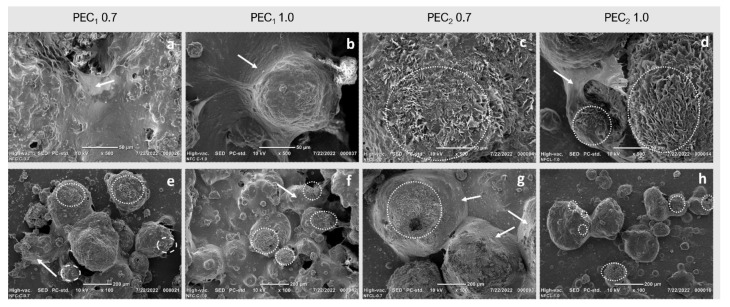
SEM images of dehydrated cocoa butter water emulsions stabilized by C_1_ and C_2_ at 0.7 wt %, 1.0 wt % taken at (**a**–**d**) 500× and (**e**–**h**) 100×. Dashed circles indicate multidirectional crystal growth, and white arrows indicate the three-dimensional network formed by CNFs around fat droplets after high-shear mixing.

**Figure 8 polymers-15-04157-f008:**
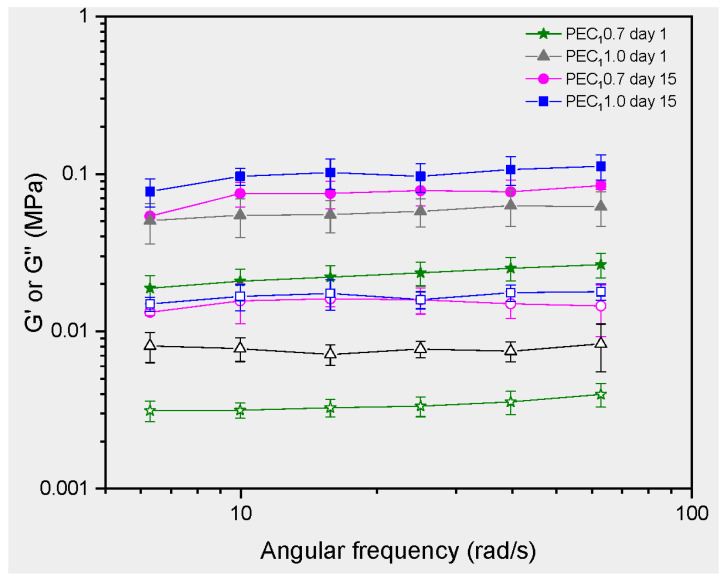
Rheology tests in PEs. Frequency sweep, with G′ represented by filled symbols and G″ by empty symbols.

**Table 1 polymers-15-04157-t001:** Description of labels for PEs.

Sample	Description
PEC_1_ 0.7	PEs stabilized with 0.7 wt % of C_1_ CNFs
PEC_1_ 1.0	PEs stabilized with 1.0 wt % of C_1_ CNFs
PEC_2_ 0.7	PEs with 0.7 wt % of C_2_ CNFs
PEC_2_ 1.0	PEs stabilized with 1.0 wt % of C_2_ CNFs

**Table 2 polymers-15-04157-t002:** Properties of CNFs. Water, organic, and total extractables of CNFs. Polymerization degree (DP).

Sample	Water Extractables (%)	Organic Extractables (%)	Total Extraction (%)	DP
Cocoa shell	25.52 ± 2.74	27.68 ± 2.96	53.20 ± 2.74	N/A
C_1_	10.74 ± 1.66	11.80 ± 1.9	22.53 ± 3.53	1098.52 ± 1.10
C_2_	4.86 ± 0.14	5.66 ± 0.22	10.53 ± 0.32	554.31 ± 1.67

**Table 3 polymers-15-04157-t003:** Characteristic bands in ATR–FTIR spectra of cocoa shell nanocellulose and their assignments according to the literature.

Wavenumber (cm^−1^)	Absorbances (a.u.)		Description	References
	C_1_	C_2_		
3405	1.1510	0.7503	O(2)H⋯O(5) intermolecular bonding	[62]
3335	1.5017	0.8003	Intramolecular contribution	
3350	1.2120	0.8033	OH stretching (intramolecular hydrogen bonds)	[61]
3285	1.1282	0.7271	O(6)H⋯O(3) intermolecular bonding	[61]
3305	1.1565	0.7534	OH stretching (intermolecular hydrogenbonds in the 101 plane)	[61,62]

## Data Availability

The raw data for the FTIR spectra and rheological analysis is available as Appendix A.

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
