# Peer review of "Nanocellulose from Cocoa Shell in Pickering Emulsions of Cocoa Butter in Water: Effect of Isolation and Concentration on Its Stability and Rheological Properties"

_polymers, 2023, doi:10.3390/polym15204157_

Round 1
Reviewer 1 Report
Comments:
The authors have embarked on a critical examination of the rheological properties of emulsions, particularly looking at the interfacial interaction between cellulosic fibers and cocoa fat droplets. Their research illuminates the characteristics of the gel-like emulsions and the impact of different concentrations of CNFs on the behavior of these emulsions. While the manuscript provides a substantive amount of data and observations, there are specific areas where more clarity, additional data, or a more detailed discussion would further strengthen the study and its findings. A major revision is recommended.
- The title should reflect the emphasis on CNFs' concentration and its effect on rheological properties more explicitly.
- The introduction lacks a comprehensive review of literature pertaining to the methods and tools used in studying emulsions. Could you add more prior work to set the context better?
- The authors mention that "PEC:0.7 and PEC1.0 can be classified as gel-like emulsions." What criteria are being used to classify emulsions as "gel-like"?
- In Figure 8, consider adding more descriptive legends or captions to help readers understand the graphical data presented better.
- When discussing the storage modulus G', it would be helpful to provide more insight into why its independence from frequency is significant.
- Clarify the difference between PEC0.7 and PEC.1.0. How were these ratios determined?
- In your methods section, can you provide more explicit details about the high shear mixing processing and its parameters?
- Please elucidate more on the importance and role of the three-dimensional network formed by CNFs around droplets.
- Expand on how the three-dimensional network contributes to the observed elastic behavior in the emulsion.
- Could you elaborate on how CNFs from banana rachis differ or are similar to the CNFs used in this study?
- The comparison with Velásquez-Cock et al. (2021) lacks a direct correlation with the current study. Could you clarify the relevance of mentioning their findings?
- The discussion about crystallization of cocoa butter-in-water emulsions and its relevance needs expansion. How do the various crystallization forms of cocoa butter affect the emulsions?
- It would be beneficial to readers if you provide more background on Pickering emulsions and their significance in the context of the study.
- Clarify the processes and importance of ATR-FTIR spectra and its correlation with the research findings.
- There's mention of a "second alkaline treatment accompanied by a treatment with HCI in C." Can you expand on what this treatment entails and its implications?
- The AFM results are referenced, but the details and significance are not clear. Could the authors provide more context or a more in-depth discussion on this?
- Explain why a "lower degree of polymerization" in nanocellulose is significant for the study.
- The conclusion states that the emulsions developed offer a possibility to reduce the content of cocoa butter in food and cosmetic formulations. Could you provide potential practical applications or commercial implications?
- I recommend including a future work section. What are the next steps or further studies that can be undertaken based on these findings?
- A more robust discussion on the broader implications of these findings in the world of food and cosmetics could be beneficial.

None.
Author Response
|
Comment |
Answer |
||||||||||
|
The title should reflect the emphasis on CNFs' concentration and its effect on rheological properties more explicitly. |
Thank you for your comment; the title has been modified: Nanocellulose from Cocoa Shell in Pickering Emulsions of Cocoa Butter in Water: Effect of Isolation and Concentration on its Stability and Rheological Properties. |
||||||||||
|
The introduction lacks a comprehensive review of literature pertaining to the methods and tools used in studying emulsions. Could you add more prior work to set the context better?
|
Thank you for your comment. We have rewritten the introduction and included literature on Pickering Emulsions (PEs), methods, and tools used in PES characterization to provide a better context. Changes in introduction are highlighted in red
|
||||||||||
|
The authors mention that "PEC:0.7 and PEC1.0 can be classified as gel-like emulsions." What criteria are being used to classify emulsions as "gel-like"? |
Thank you for your comment. A discussion about the criteria used to classify emulsions as gel-like was included in lines 512 to 515. … Results for the viscoelastic properties of the emulsion allowed to establish that PEC10,7 and PEC11.0 can be classified as gel-like emulsions, as the storage modulus G′ showed a higher value than the loss modulus G″, and G’ is independent of frequency [22], [93], [94]. The elastic behavior dominates over the viscous one for the evaluated PEs, as the value of G’ is greater than G’’ through the evaluated frequency range [22], [93]….. |
||||||||||
|
In Figure 8, consider adding more descriptive legends or captions to help readers understand the graphical data presented better.
|
Thank you for your comment. The figure caption has been improved.
|
||||||||||
|
When discussing the storage modulus G', it would be helpful to provide more insight into why its independence from frequency is significant. |
Thank you for your comment. A discussion about why the independence of G' from frequency is relevant was included in lines 516-526 …Cocoa butter in water Pickering emulsions (4:1), were prepared using a method based on previous works [29], [35], [36]. Cocoa butter was melted for one hour at 110 °C to remove all crystal memory [24]. Subsequently, as illustrated in Figure 1, nanocellulose suspension, either C1 or C2 at two different concentrations (0.7 wt.% and 1.0 wt.%) [22], [36], was added to the molten cocoa butter at room temperature (25 °C). Finally, the two-phase system was processed through a high shear mixing (HSM) or high-speed rotor-stator system (Ultraturrax T50, IKA, Cologne, Germany, impeller model S25N-25F) at 25,000 rpm for 180 s. HSM is based on a rotor-stator system, where the rotor can rotate at 25,000 rpm and is enclosed in a stator with a gap between them ranging from 100 to 3000 µm. This equipment applies high shear force, which disperses fat droplets in the water phase. After HSM, the samples were labeled PEC1 0.7, PEC1 1.0, PEC2 0.7, and PEC2 1.0. They were kept in a container for transportation. Table 1 describes the sample labels for PEs.
Table 1. description of labels for PEs
|
||||||||||
|
Clarify the difference between PEC0.7 and PEC.1.0. How were these ratios determined?
|
Thank you for your comment; we have included information on the methodology to better understand the differences in samples. Additionally, we have included some references to clarify how CNF ratios were determined. See lines 149-163.
|
||||||||||
|
In your methods section, can you provide more explicit details about the high shear mixing processing and its parameters? |
Thank you for your comment information about shear mixing was included in lines 151 to 156
….Finally, the two-phase system was processed through a high shear mixing (HSM) or high-speed rotor-stator system (Ultraturrax T50, IKA, Cologne, Germany, impeller model S25N-25F) at 25,000 rpm for 180 s. HSM is based on a rotor-stator system, where the rotor can rotate at 25,000 rpm and is enclosed in a stator with a gap between them ranging from 100 to 3000 µm. This equipment applies high shear force, which disperses fat droplets in the water phase. After HSM, the samples were labeled PEC1 0.7, PEC1 1.0, PEC2 0.7, and PEC2 1.0. They were kept in a container for transportation. Table 1 describes the sample labels for PEs.… |
||||||||||
|
Please elucidate more on the importance and role of the three-dimensional network formed by CNFs around droplets
Expand on how the three-dimensional network contributes to the observed elastic behavior in the emulsion.
|
Thank you for your comment. A discussion about the importance of the three-dimensional network and its relation whit elastic behaviour of emulsion was included in lines: 324 – 343, 526-536
...Consequently, the isolation treatment influences the chemical composition of cellulose and its fibrillation [68]. Figure 3a-f shows that both samples C1 and C2 were nanofibrillated and presented an entangled network of cellulosic material. The formation of this three-dimensional entangled network is relevant in PEs formulation, because it favors the irreversible adsorption of nanocelluloses at the oil-water interface, increasing the steric hindrance between the droplets and inhibits the free movement of the droplets, resulting in a lower coalescence rate [69]. For this reason, differences between the entangled networks formed by C1 and C2 should be comprehended with the aim of understanding the properties of PEs. Figures 3 a-c present the morphology of C1 at different scales, AFM images showed individual nanofibers with diameters around 30.5 ± 12.5 nm, and the presence of fiber bundles and sub-fibrillated cellulosic material with diameters ranging between 275 to 380 nm. In addition, Figures 3 d-f show the morphology of C2. It is evident the presence of separated individual nanofibers with diameters around 26.9 ± 8.2 nm and fiber bundles with diameters ranging between 98 to 225 nm. AFM images reveal the presence of bundles in both samples C1 and C2, marked with white arrows in Figures 3b-c and Figures 3 e-f, respectively. In summary, AFM revealed that C1 shows an entangled network with lower degree of fibrillation, which is associated with its isolation treatment [34]. C1 CNFs also showed the presence of hemicellulose, which is related to a lower degree of fibrillation and bundles with larger diameters [68]…….
… As observed in microscopic, Figure 5, and SEM images, Figure 7, CNFs formed a three-dimensional network around droplets after high shear mixing, which plays a prominent role in stability and rheological behavior [95]. As observed in Figure 7 a-c and b-d, nanocellulose was absorbed over the surface of the oil droplets to form a covering layer after the high shear mixing processing, resulting in a stable droplet-fiber 3D network structure. This network structure contributes to the elasticity of the emulsion. CNFs network can deform elastically when subjected to stress and then return to their original shape when the stress is removed…… |
||||||||||
|
Could you elaborate on how CNFs from banana rachis differ or are similar to the CNFs used in this study? |
Thank you for your comment a discussion about relation between CNFs from cocoa shell and banana rachis was included in lines 540 -552.
…Velásquez-Cock et al. (2021) extracted CNFs from banana rachis following the same procedure as that used for C2 and evaluated the effect of CNFs concentration on rheological properties of coconut or palm oil-in-water based Pickering emulsions [22]. They reported an increase in G’ values around three orders of magnitude when CNFs concentration increased from 0.15 wt.% to 0.7 wt.%. Another study added CNFs from banana rachis, isolated using C2 methodology, to Curcuma longa suspensions reported similar results. Increasing the CNFs concentration from 0.1 wt.% of CNFs to 0.9 wt.% enhanced G’ by nearly 3 orders of magnitude and improved the stability over 30 days. [93]. Banana rachis and cocoa husk are both agro-industrial by-products. Despite using the same CNFs isolation methodology, the resulting CNFs exhibit different physicochemical characteristics, mainly due to variations in their physicochemical properties [10], [33]. These differences necessitated different concentrations for stabilizing PEs, as is the case observed for C1 and C2 CNFs….
|
||||||||||
|
The comparison with Velásquez-Cock et al. (2021) lacks a direct correlation with the current study. Could you clarify the relevance of mentioning their findings?
|
Thank you for your comment. A discussion about the relevance of the comparison between this study and Velásquez-Cock et al (2021) was included in lines 540 -552.
…Velásquez-Cock et al. (2021) extracted CNFs from banana rachis following the same procedure as that used for C2 and evaluated the effect of CNFs concentration on rheological properties of coconut or palm oil-in-water based Pickering emulsions [22]. They reported an increase in G’ values around three orders of magnitude when CNFs concentration increased from 0.15 wt.% to 0.7 wt.%. Another study added CNFs from banana rachis, isolated using C2 methodology, to Curcuma longa suspensions reported similar results. Increasing the CNFs concentration from 0.1 wt.% of CNFs to 0.9 wt.% enhanced G’ by nearly 3 orders of magnitude and improved the stability over 30 days. [93]. Banana rachis and cocoa husk are both agro-industrial by-products. Despite using the same CNFs isolation methodology, the resulting CNFs exhibit different physicochemical characteristics, mainly due to variations in their physicochemical properties [10], [33]. These differences necessitated different concentrations for stabilizing PEs, as is the case observed for C1 and C2 CNFs….. |
||||||||||
|
The discussion about crystallization of cocoa butter-in-water emulsions and its relevance needs expansion. How do the various crystallization forms of cocoa butter affect the emulsions?
|
Thank you for your comment. A discussion about its relevance was included in lines 556-571.
…Cocoa butter is a complex fat known to crystallize in six different polymorphs [97]. These various forms of cocoa butter crystallization can significantly impact the properties of emulsions formulated with this ingredient. For example, the different crystalline forms have distinct melting points, with the α polymorph being less stable than the β form, which typically has a higher melting point [98]. Therefore, if cocoa butter in an emulsion undergoes a phase transition from a more stable to a less stable form during storage or processing, it can lead to undesirable changes in texture, appearance, and product deterioration [98]. Furthermore, the crystallization of solid fats plays a pivotal role in the partial coalescence of droplets in oil-in-water (O/W) emulsions [98]. This type of destabilization is of particular interest in food applications and is a prerequisite for producing whipped cream and ice cream [98], [99]. Additionally, the crystallization behavior of these fats in emulsions is intricate and subject to various influences, such as droplet size, collision dynamics, additives, emulsifiers, crystallization medium, and thermal fluctuations during storage [98], [100]. It has also been reported that shear forces can impact the crystallization of fats in emulsions and promote the formation of higher polymorphs [101]… |
||||||||||
|
It would be beneficial to readers if you provide more background on Pickering emulsions and their significance in the context of the study.
|
Thank you for your comment. The introduction has been rewritten, and literature about Pickering Emulsions (PEs) and their significance in the context of the study was included. Changes in introduction are higlihted in red |
||||||||||
|
Clarify the processes and importance of ATR-FTIR spectra and its correlation with the research findings.
|
Thank you for your comment, a discussion about and importance of ATR-FTIR spectra and its correlation with the research findings, was included in lines 233 to 245:
… CNFs isolated from cocoa shells will be evaluated for the development of cocoa butter-in-water Pickering emulsions. These emulsions stabilize as the CNFs create a physical barrier and a network, acting as an obstacle between contiguous droplets, preventing droplet coalescence [44]. Factors influencing the stability of such emulsions include morphology, size and shape of the CNFs, as well as their chemical composition, which can affect the surface charge and interactions. The morphology and chemical composition of CNFs are closely linked to their isolation process [44]. Therefore, the chemical composition of C1 and C2 CNFs was assessed using various techniques, including ATR-FTIR, fluorescence microscopy, and soluble extractives. Surface hydroxyl groups in CNFs make them hydrophilic polymers capable of stabilizing oil-in-water (O/W) emulsions [44]. ATR-FTIR is a commonly used technique for studying these groups and their interactions with both cellulosic and non-cellulosic components present in CNFs [34]…..
|
||||||||||
|
There's mention of a "second alkaline treatment accompanied by a treatment with HCI in C." Can you expand on what this treatment entails and its implications?
|
Thank you for your coment. A discussion about The second alkaline treatment was included in lines 310 to 322
…Previous studies on banana rachis and cocoa husk have proposed the use of an alkaline four sequence to isolate C2 CNFs [10], [33], described in Figure 1. This sequence is based on the cleavage of β-aryl-O-4 linkages of lignin by alkaline media, exposing new zones for the following treatment [63], the degradation of phenolic and non-phenolic structures in lignin by chloride dioxide [64]. After the elimination of lignin, a secondary alkaline treatment can react with muconic acids produced during delignification, while using the freshly exposed surfaces to cleave the hemicellulose-cellulose bonds present, reducing the presence of hemicelluloses on the final structure [65]. Finally, a demineralization treatment with HCl at a low concentration was used, acid concentration allowed a reduction of the cellulosic hydrolysis [33]… |
||||||||||
|
The AFM results are referenced, but the details and significance are not clear. Could the authors provide more context or a more in-depth discussion on this?
|
Thank you for your comment. The discussion in AFM was improved in lines 323 – 343.
…Consequently, the isolation treatment influences the chemical composition of cellulose and its fibrillation [68]. Figure 3a-f shows that both samples C1 and C2 were nanofibrillated and presented an entangled network of cellulosic material. The formation of this three-dimensional entangled network is relevant in PEs formulation, because it favors the irreversible adsorption of nanocelluloses at the oil-water interface, increasing the steric hindrance between the droplets and inhibits the free movement of the droplets, resulting in a lower coalescence rate [69]. For this reason, differences between the entangled networks formed by C1 and C2 should be comprehended with the aim of understanding the properties of PEs. Figures 3 a-c present the morphology of C1 at different scales, AFM images showed individual nanofibers with diameters around 30.5 ± 12.5 nm, and the presence of fiber bundles and sub-fibrillated cellulosic material with diameters ranging between 275 to 380 nm. In addition, Figures 3 d-f show the morphology of C2. It is evident the presence of separated individual nanofibers with diameters around 26.9 ± 8.2 nm and fiber bundles with diameters ranging between 98 to 225 nm. AFM images reveal the presence of bundles in both samples C1 and C2, marked with white arrows in Figures 3b-c and Figures 3 e-f, respectively. In summary, AFM revealed that C1 shows an entangled network with lower degree of fibrillation, which is associated with its isolation treatment [34]. C1 CNFs also showed the presence of hemicellulose, which is related to a lower degree of fibrillation and bundles with larger diameters [68]…
|
||||||||||
|
Explain why a "lower degree of polymerization" in nanocellulose is significant for the study.
|
Thank you for your comment. The importance of polymerization degree for this study in lines 347 to 373.
…On the other hand, the isolation process influences the length of CNFs [34], [70].. Zuluaga et al (2009), evidenced that a reaction whit HCl 2 M longitudinally cuts the cellulose microfibrils. This is typical of CNFs treated with strong acids that preferentially degrade the disordered regions along the microfibrils [33]. The length and polymerization degree of CNFs are interconnected parameters that cannot be changed independently easily [68]. A higher polymerization degree (DP) is directly related to longer fibers [39], [41], [43], and fiber length is a parameter that directly affects the stability of PEs. It is expected that long fibers aggregates are able to form an entangled network, which would increase the viscosity of the formed suspension, compared to fibers with a lower aspect ratio [41], [71]. In CNFs, the DP is measured through its intrinsic viscosity, which is a characteristic of macromolecules that is directly related to their ability to disturb flow and indirectly relates to the chain length. The relation between intrinsic viscosity and polymerization degree for cellulose was established in the XXth century by different authors [38], and the constants used by Marx-Figini 1978 were used to determine the DP reported in Table 1 [38]. The DP obtained for C1 and C2 was 1098.52 ± 1.10 and 554.31 ± 1.67, respectively, which are within the range reported in the literature [34], [72]–[74]. The DP of a cellulosic polymer can be around 10,000 in wood cellulose but, after its degradation and purification process, the DP is reduced to about 300–1700 [72]. In cellulose from agroindustrial by-products like banana rachis, DP was reported between 1012 and 612, depending on homogenization process [34]. Another aspect that influences the DP is the presence of non-cellulosic components. It was reported that a higher concentration of non-cellulosic components is related with a higher DP, as was observed for commercial [73] and cotton cellulose [74]. Finally, HCl used to remove mineral traces in C2 [33] also reduces the polymerization degree, as it degrades the short, disordered, non-crystalline regions in cellulose [74], [75]. Therefore, the higher DP observed for C1 is manly associated with a higher presence of non-cellulosic components and a lack of a mild acid hydrolysis present in C2 sample… |
||||||||||
|
The conclusion states that the emulsions developed offer a possibility to reduce the content of cocoa butter in food and cosmetic formulations. Could you provide potential practical applications or commercial implications? |
Thank you for your insightful comment. We have incorporated your input into our discussion, which now covers the potential practical applications and implications of these findings within the cosmetic and food industries. Additionally, we have included a future work section that focuses on cocoa-butter in water-based polymeric emulsions (PEs). Your contribution has greatly enriched our research. See lines: 577- 591. Therefore, the findings of this work are significant from a scientific perspective and have implications for various applications, including food, cosmetics, and pharmaceutical formulation. Some potential directions and applications stemming from this research are as follows: The use of cocoa butter-in-water Pickering emulsions in the food industry offers several key advantages: reduced caloric density of products [102], the potential for incorporating active ingredients to enhance nutritional value [103], and lowered production costs due to reduced raw material usage [97]. However, further research is required in this field, particularly concerning the crystallization of cocoa butter with the integration of new active compounds. The formulation of innovative aqueous-based products, such as nano-delivery systems and edible coatings, can be developed using stable cocoa butter-in-water emulsions. This approach leverages the antioxidant, moisturizing, and barrier properties of cocoa butter, which is widely utilized in the food, pharmaceutical, and cosmetic industries [104]. |
||||||||||
|
I recommend including a future work section. What are the next steps or further studies that can be undertaken based on these findings?
|
Thank you for your insightful comment. We have incorporated your input into our discussion, which now covers the potential practical applications and implications of these findings within the cosmetic and food industries. Additionally, we have included a future work section that focuses on cocoa-butter in water-based polymeric emulsions (PEs). Your contribution has greatly enriched our research. See lines: 577- 591.
Therefore, the findings of this work are significant from a scientific perspective and have implications for various applications, including food, cosmetics, and pharmaceutical formulation. Some potential directions and applications stemming from this research are as follows: The use of cocoa butter-in-water Pickering emulsions in the food industry offers several key advantages: reduced caloric density of products [102], the potential for incorporating active ingredients to enhance nutritional value [103], and lowered production costs due to reduced raw material usage [97]. However, further research is required in this field, particularly concerning the crystallization of cocoa butter with the integration of new active compounds. The formulation of innovative aqueous-based products, such as nano-delivery systems and edible coatings, can be developed using stable cocoa butter-in-water emulsions. This approach leverages the antioxidant, moisturizing, and barrier properties of cocoa butter, which is widely utilized in the food, pharmaceutical, and cosmetic industries [104]. |
||||||||||
|
A more robust discussion on the broader implications of these findings in the world of food and cosmetics could be beneficial.
|
Thank you for your insightful comment. We have incorporated your input into our discussion, which now covers the potential practical applications and implications of these findings within the cosmetic and food industries. Additionally, we have included a future work section that focuses on cocoa-butter in water-based polymeric emulsions (PEs). Your contribution has greatly enriched our research. See lines: 577- 591.
Therefore, the findings of this work are significant from a scientific perspective and have implications for various applications, including food, cosmetics, and pharmaceutical formulation. Some potential directions and applications stemming from this research are as follows: The use of cocoa butter-in-water Pickering emulsions in the food industry offers several key advantages: reduced caloric density of products [102], the potential for incorporating active ingredients to enhance nutritional value [103], and lowered production costs due to reduced raw material usage [97]. However, further research is required in this field, particularly concerning the crystallization of cocoa butter with the integration of new active compounds. The formulation of innovative aqueous-based products, such as nano-delivery systems and edible coatings, can be developed using stable cocoa butter-in-water emulsions. This approach leverages the antioxidant, moisturizing, and barrier properties of cocoa butter, which is widely utilized in the food, pharmaceutical, and cosmetic industries [104]. |
Reviewer 2 Report
The authors studied the preparation and characterization of cocoa butter emulsions stabilized by nanocellulose from cocoa shells. The study is of interest to the food industry and complies with sustainable development goals. However, many grammatical and language errors need to be addressed. Some sentences are not understandable. Some letters are not in English (line 56, reduce de content). The manuscript should be revised and edited by an expert proofreader without affecting the scientific soundness of the manuscript.
Other comments,
- many abbreviations were mentioned for the first time in the manuscript without explaining their meaning; i.e, DP, AFM, GRAS, EFSA, FDA, etc.
- Statistical analysis details are not mentioned in the manuscript. Information about the the number of replications, statistical tests and software should be written.
- Concise In-depth discussions are needed in the discussion section.
-
Many grammatical and language errors need to be addressed. The whole manuscript should be edited very carefully.
Author Response
|
Comment |
Answer |
|
The authors studied the preparation and characterization of cocoa butter emulsions stabilized bynanocellulose from cocoashells. The study is of interest to the food industry and complies with sustainable development goals. However, many grammatical and language errors need to be addressed. Some sentences are not understandable. Some letters are not in English (line 56, reducedecontent). The manuscript should be revised and edited by an expert proofreader without affecting the scientific soundness of the manuscript. |
Thank you for your insightful comment. We have incorporated your inputs into our manuscript.
The manuscript was checked by a person fluent in English writing. All changes in the text are highlighted in red. |
|
-Many abbreviations were mentioned for the first time in the manuscript without explaining their meaning; i.e, DP, AFM, GRAS, EFSA, FDA, etc |
All abreviatures were described in text. Changes in text are highlighted in red. |
|
-Statistical analysis details are not mentioned in the manuscript Information about the the number of replications, statistical tests and software should bewritten |
Thank you for your comment. A section with the statistical analysis was included in methodology. See lines: 249-259 |
|
-Concise In-depth discussions are needed in the discussion section. |
Thank you for your insightful comment. We have significantly improved the discussion section. All changes in the text are highlighted in red. |
Reviewer 3 Report
Dear Authors,
the submitted manuscript I have reviewed is very chaotic and contains numerous technical issues. I should have rejected this work. However, to acknowledge your effort and experimental work you did to obtain the results, I suggest the major revision. It is not because of the scientific soundness, but the technical aspects. Please, in the future, prepare the manuscripts of better quality to be able to share your results with the international audience. Practice makes perfect.
I am very sorry for the technical side of my review report. The Journal's template does not allow for putting bullet points.
-------------
The submitted paper describes the use of nanoparticles from cocoa shell as stabilizers for cocoa butter – in – water Pickering emulsions. The Authors present the extended characterization of nanoparticles and emulsions using Fourier spectroscopy, microscopy and rheological analysis. They show that the obtained emulsions were stable against coalescence and they consider also so-called arrested (partial) coalescence of the droplets. The topic could be interesting for the readers of Polymers after a thorough revision. My specific comments to the manuscript are attached below.
1. In the Abstract, the Authors mentioned ”limited coalescence”. However, they did not use this term even in the Introduction (see, e.g., line 68). It should be improved. Moreover, I did not find the reference to this term in the discussion of the obtained results.
2. Figure 1 is not precisely described in the caption. Please, extend the caption.
3. The acronyms and symbols in the manuscript should be explained. See, e.g., OMNIC 9 and TAPPI method (line 172 and 176), I_beta (line 277), and CS (line 311).
4. What were the parameters of sonication of the samples for AMF measurements (line 182-184)? How could the used parameters affect the sample morphology under AMF?
5. The Authors said that the procedure of rheological measurements for PEs was the same as for nanocellulose particles. What about the stability of PEs under high shear rates? Was it controlled during the process?
6. The Section 2.4.5. is repeated. It should not have happened for the manuscript submitted to the international journal of a high IF. Please, put more effort in the future to the technical preparation of the manuscript.
7. The Authors did their job characterizing cellulose particles. However, the whole description (from line 261 to 410) is lacking the aim of the characterization. The Authors seem to describe the results, they relate them to the literature (that is totally fine), but – what was the goal of such a thorough characterization? The inclusion of subsection in the Chapter 3 could help in better “flow” of the text of the manuscript.
8. There is no a direct bridge between the very extended description of the results from testing of nanoparticles and PEs (see, line 470-471). It should be better linked.
9. What was the reason why there are no data for higher concentration of the PEs after 15 days (Figure 6)? It should be explained in the text of the manuscript.
The language of the manuscript is rather understandable after corrections. However, the careful proofreading of the manuscript (and supplementary materials) is still recommended as there are many linguistic and technical issues in the text, as shown below. It is strange taking into account that there is a revised version of the manuscript.
- - The affiliation of the Authors does not contain important information on the University. In Supplementary Materials, it is surprisingly correct.
- - In the Abstract, “strategies” is in bold. Why?
- - In the line 19, “d” refers to days, but it is not clear that is why it should be extended as “days”. The same happens in other part of the manuscript (e.g., line 187, 218, etc.).
- - In the Introduction, there are additional spaces between paragraphs (empty lines, e.g., 75).
- - What do the Authors mean in the line 115 by “its goal”. Whose goal? Please, stress out.
- - Why is the company name in italics in the line 126?
- -The reference brackets should be corrected. See, e.g., line 134 and 137, 150 and 152, 207 and 226, 284, and 286, 351, etc.. Why are the references in separated brackets? Please, look carefully at the technical requirements of the MDPI Journals. For instance, the names of company, the city and the country should be provided in Materials and Methods. It is not, e.g., for the line 140 and 167.
- - The sentence in the line 145 is not understandable.
- - In the line 162, the caption should start with capital letter.
- - It is not clear why the viscosity symbol in the Eq. 1 is in []. The units of the quantities should be provided.
- - There should be spaces corrected in line 200. In the same line there is “obj”. What is this?
- - From technical point, please, note the differences in paragraphs width (for instance, the difference between line 235 and 238). There are also different font styles and sizes (see, e.g., References).
- - There is a typo in the name of the US in the line 205.
- - Please, verify the 0.01% starting amplitude in the line 209.
- - Figure 1 and Figure 2 in Supplementary Information should be cited correctly (line 307 and 328).
- - There is a redundant space in the line 310.
- - Please, extend the caption of Table 2.
- - Problems with spaces and commas, and other issues. It should be reedited. The same concerns the added paragraph from the line 376.
- - The grammar of the line 365 should be corrected.
- - 20th century instead of XXth century (line 388).
- - The authors of the original paper from 1978 are wrongly provided. Why is “1978” used in line 389? Similarly, there is a date in the line 519, 524, and 568. Why? It is not within the reference style.
- - The graphs presenting viscosity data (Figure 4) should be of much better quality. In this form, these do not show the differences in symbols used for G’ and G’’, for instance. The caption of Figure 4 is misleading. It does not present a test. It presents the results from measurements.
- - The whole paragraph starting from line 445 should be revised. I do not understand why the same set of references (with a wrong style, as above) appears three times and why there are so many references?
- - “Cremation” should rather be replaced by “creaming” (“cremated” by “creamed”), line 465.
- - In the line 486, “Day 15” seems to be not grammatically correct.
- - In the line 511, there is a dot before [citation]. It should be corrected.
- - There are some technical issues in the caption of Figure 8. Please, edit this.
- - The “:” in the line 608 suggests that the following statements should be in bullet points.
-
I
The quality of English is fine. I listed the issues I found in my report. However, the overal technical side of the paper has to be improved.
Author Response
Dear Reviewer, on behalf of our team, we want to thank for your comments to improve this paper so it can be published in Polymers. All observations were made and answered point by point, as shown in the Table attached.
|
Reviewer comments |
Answer |
|
In the Abstract, the Authors mentioned ”limited coalescence”. However, they did not use this term even in the Introduction (see, e.g., line 68). It should be improved. Moreover, I did not find the reference to this term in the discussion of the obtained results. |
Authors did not identify the term “limited coalescence”, in abstract or any other part of text. |
|
Figure 1 is not precisely described in the caption. Please, extend the caption |
Thank you for your comment. The figure caption has been updated |
|
The acronyms and symbols in the manuscript should be explained. See, e.g., OMNIC 9 and TAPPI method (line 172 and 176), I_beta (line 277), and CS (line 311). |
Thank you for your comment. The acronyms and symbols have been explained in the manuscript text:
OMNIC: This is the name of software developed by Thermo Fisher Scientific, which was used for data analysis and processing in various scientific fields, including spectroscopy and microscopy. A clarification for this was included in the text, specifically on line 174.
TAPPI: Information about this acronym is provided in lines 128.
Iβ: This symbol refers to a specific type of crystal structure or molecular arrangement, particularly in the context of chemistry and materials science. Cellulose, can exist in various crystalline forms, with "Iβ" denoting one of these forms. The authors believe that clarification of this symbol in the text is unnecessary.
CS: Information about this acronym is also included in lines 128." |
|
What were the parameters of sonication of the samples for AMF measurements (line 182-184)? How could the used parameters affect the sample morphology under AMF? |
Thank you for your comment. Sonication parameters were included in line 187.
As you mentioned, sonication parameters have an effect on the final morphology of CNFs observed under AFM. Therefore, ultrasonic amplitude, sonication time, frequency, and the type of sonication were selected according to the literature to minimize the impact of ultrasound on CNFs morphology. |
|
The Authors said that the procedure of rheological measurements for PEs was the same as for nanocellulose particles. What about the stability of PEs under high shear rates? Was it controlled during the process? |
The viscoelastic properties of the PE were assessed by a frequency sweep in the lineal viscoelastic region, which does not exert irreversible changes in the rheological properties of the sample (Zhu et al.,2020). While the sweep evidence the effect of different frequencies in the elastic and loss moduli (G' and G'') and the complex viscosity of the samples.
Zhu, Y., Gao, H., Liu, W., Zou, L., & McClements, D. J. (2020). A review of the rheological properties of dilute and concentrated food emulsions. Journal of Texture Studies, 51(1), 45-55. |
|
The Section 2.4.5. is repeated. It should not have happened for the manuscript submitted to the international journal of a high IF. Please, put more effort in the future to the technical preparation of the manuscript. |
Thank you for your comment. Repeated information was removed from the text. |
|
The Authors did their job characterizing cellulose particles. However, the whole description (from line 261 to 410) is lacking the aim of the characterization. The Authors seem to describe the results, they relate them to the literature (that is totally fine), but – what was the goal of such a thorough characterization?
The inclusion of subsection in the Chapter 3 could help in better “flow” of the text of the manuscript. |
Authors consider that the aim of CNFs characterization is well described between lines 264 to 273: “…CNFs isolated from cocoa shells will be evaluated for the development of cocoa butter-in-water Pickering emulsions. These emulsions stabilize as the CNFs create a physical barrier and a network, acting as an obstacle between contiguous droplets, preventing droplet coalescence [44]. Factors influencing the stability of such emulsions include morphology, size and shape of the CNFs, as well as their chemical composition, which can affect the surface charge and interactions. The morphology and chemical composition of CNFs are closely linked to their isolation process…” However, an additional text was included below line 273 to emphasize the idea presented in this paragraph: Hence, an assessment was conducted on the chemical, physical, and morphological characteristics of both CNFs, C1 and C2, to comprehend the impact of the isolation process on the physical barrier and network responsible for stabilizing the emulsion. Subsections were included in chapter 3. |
|
There is no a direct bridge between the very extended description of the results from testing of nanoparticles and PEs (see, line 470-471). It should be better linked. |
Thank you for comment a link between characterization of CNFs and characterization of PEs have been included, between lines 452-454: For this reason, two different concentrations of both types of CNFs, 0.7 wt% and 1.0 wt%, were evaluated to develop PE with cocoa butter. The following section discusses the effects of isolation and cellulose concentration on PE. |
|
What was the reason why there are no data for higher concentration of the PEs after 15 days (Figure 6)? It should be explained in the text of the manuscript.
|
The study of emulsion stability in PE with higher concentrations of CNFs is beyond the scope of this paper. The evaluated concentrations of CNFs were defined according to the literature, as reported in methodology, line 155. The study of emulsion stability after day 15 was also beyond the scope of the paper. A higher concentration of CNFs increases viscosity and requires the use of other techniques to process PEs. However, since you mentioned Figure 6, we are not sure if you are referring to the absence of microscopy images for PEC20.7 and PEC21.0 on day 15. In that case, the reason is described in the text in lines 502 to 505: “Samples formulated with C2 (PEC2 0.7 and PEC2 1.0) did not show satisfactory stability before 15 days, as droplet agglomerates were observable to the naked eye, as marked with white arrows on Figure 5c. As their size was larger than the microscope focus, these images were not included in Figure 6.” |
|
The language of the manuscript is rather understandable after corrections. However, the careful proofreading of the manuscript (and supplementary materials) is still recommended as there are many linguistic and technical issues in the text, as shown below. It is strange taking into account that there is a revised version of the manuscript. |
Thank you for your comment. The manuscript has been carefully reviewed, and both linguistic and technical issues have been addressed. |
|
The affiliation of the Authors does not contain important information on the University. In Supplementary Materials, it is surprisingly correct. |
Thank you for your comment. The authors' affiliation has been updated to ensure that both the manuscript and the supplementary materials contain consistent information |
|
In the Abstract, “strategies” is in bold. Why? |
Thank you for your comment. The use of bold in this word has been corrected |
|
The line 19, “d” refers to days, but it is not clear that is why it should be extended as “days”. The same happens in other part of the manuscript (e.g., line 187, 218, etc.). |
Thank you for your comment. The mistake has been corrected, and the word 'day' has been made consistent throughout the entire manuscript |
|
In the Introduction, there are additional spaces between paragraphs (empty lines, e.g., 75). |
Thank you for your comment. the mistake has been corrected |
|
What do the Authors mean in the line 115 by “its goal”. Whose goal? Please, stress out. |
Thank you for your comment. A grammatical correction has been made: …To achieve these objectives, two different chemical treatments were employed to isolate nanocellulose from cocoa shells. |
|
Why is the company name in italics in the line 126? |
In this case, italics were used as a convention to highlight Spanish words within the English text. |
|
The reference brackets should be corrected. See, e.g., line 134 and 137, 150 and 152, 207 and 226, 284, and 286, 351, etc.. Why are the references in separated brackets?
Please, look carefully at the technical requirements of the MDPI Journals. For instance, the names of company, the city and the country should be provided in Materials and Methods. It is not, e.g., for the line 140 and 167. |
Thank you for your comment. References have been prepared using Zotero, a bibliography software, to prevent formatting and typing errors, as well as duplicate references.
The names of the company, city, and country have been provided for all the equipment used in the Materials and Methods section |
|
The sentence in the line 145 is not understandable |
Thank you for your comment. The sentence has been rewritten: The nanocellulose suspensions C1 and C2 with concentrations of 0.1 wt.% were filtered under vacuum through a 0.2 μm nylon membrane. |
|
In the line 162, the caption should start with capital letter |
Thank you for your comment. The typo error has been corrected. |
|
It is not clear why the viscosity symbol in the Eq. 1 is in []. The units of the quantities should be provided. |
Thank you for your comment. The viscosity reported in Eq. 1 is the intrinsic viscosity, which is represented with the symbol as mentioned in line 202. The units have been included between lines 202 and 203. Units have been included between lines 202 and 203. |
|
There should be spaces corrected in line 200. In the same line there is “obj”. What is this? |
The space and the symbol  have been erased. The symbol  appears to be a placeholder or a formatting issue. It doesn't represent any specific character or meaning. |
|
From technical point, please, note the differences in paragraphs width (for instance, the difference between line 235 and 238). There are also different font styles and sizes (see, e.g., References). |
Thank you for your comments. Technical errors like spaces between paragraphs, typo and, font styles have been emended. |
|
There is a typo in the name of the US in the line 205. |
Thank you for your comment. The typo has been emended |
|
Please, verify the 0.01% starting amplitude in the line 209 |
Thank you for your comment. Amplitude |
|
Figure 1 and Figure 2 in Supplementary Information should be cited correctly (line 307 and 328). There is a redundant space in the line 310 |
Thank you for your comment. Figures in supplementary information have been cited correctly: lines 291 and 312. The typo has been emended |
|
Please, extend the caption of Table 2 |
Thank you for your comment. the table caption was extended |
|
Problems with spaces and commas, and other issues. It should be reedited. The same concerns the added paragraph from the line 376 |
Thank you for your comment. The technical mistakes have been emended |
|
The grammar of the line 365 should be corrected |
Thank you for your comment. The sentence has been corrected |
|
20th century instead of XXth century (line 388). |
Thank you for your comment. The century has been corrected |
|
The authors of the original paper from 1978 are wrongly provided. Why is “1978” used in line 389? Similarly, there is a date in the line 519, 524, and 568. Why? It is not within the reference style. |
Thank you for your comment. References have been prepared using Zotero, a bibliography software, to prevent style, and typing errors, as well as duplicate references |
|
The graphs presenting viscosity data (Figure 4) should be of much better quality. In this form, these do not show the differences in symbols used for G’ and G’’, for instance. The caption of Figure 4 is misleading. It does not present a test. It presents the results from measurements. |
Thank you for your comment. The graphs quality has been improved and, figure caption has been updated |
|
The whole paragraph starting from line 445 should be revised. I do not understand why the same set of references (with a wrong style, as above) appears three times and why there are so many references? |
Thank you for your comment. References have been prepared using Zotero, a bibliography software, to prevent style, and typing errors, as well as duplicate references |
|
Cremation” should rather be replaced by “creaming” (“cremated” by “creamed”), line 465. |
Thank you for your comment. The grammar mistakes have been emended |
|
In the line 486, “Day 15” seems to be not grammatically correct. |
Thank you for your comment. The grammar mistake has been emended |
|
In the line 511, there is a dot before [citation]. It should be corrected. |
Thank you for your comment. The technical mistake has been emended |
|
There are some technical issues in the caption of Figure 8. Please, edit this. |
Thank you for your comment. The caption has been updated |
|
The “:” in the line 608 suggests that the following statements should be in bullet points. |
Thank you for your comment. The bullet points have been added |

Round 2
Reviewer 1 Report
Comments:
The author has made good revisions and explanations to the proposed questions. The current version of the paper can now be accepted and published.
Author Response
Thank you for your comment
Reviewer 2 Report
The authors revised the manuscript carefully, especially the language. Thus, the manuscript can be accepted for publication in this journal.
Author Response
Thank you for your comment
Reviewer 3 Report
The Authors revised their manuscript according to the comments.